# Algorithm for Schroth-Curve-Type Classification of Adolescent Idiopathic Scoliosis: An Intra- and Inter-Rater Reliability Study [note 1]

**DOI:** 10.3390/children10030523

**Published:** 2023-03-08

**Authors:** Sanja Schreiber, Eric C. Parent, Gregory N. Kawchuk, Douglas M. Hedden

**Affiliations:** 1Department of Physical Therapy, University of Alberta, 8205 114 Street, 2-50 Corbett Hall, Edmonton, AB T6G 2G4, Canada; 2Department of Surgery, University of Alberta, 8440 112 Street, 2D2.24 WMC, Edmonton, AB T6G 2R7, Canada

**Keywords:** scoliosis, algorithms, classification, posture, reliability, reproducibility of results, Schroth, adolescent

## Abstract

Schroth exercises for scoliosis are prescribed based on curve types. This study aimed to determine the reliability of an algorithm for classifying Schroth curve types. Forty-four consecutive volunteers with adolescent idiopathic scoliosis, 10 to 18 years old, with curves 10° to 50°, were recruited from a scoliosis clinic. Their standing posture and Adam’s bending test were videotaped. Ten consecutive Schroth therapist volunteers from an international registry independently classified the curve types using the proposed classification algorithm. Videos were rated twice at least seven days apart. Reliability was calculated using the Gwet’s AC1 agreement coefficient for all the raters and for subgroups reporting full understanding (well-trained) and with prior algorithm experience. The intra-rater and weighted agreement coefficients for all the raters were 0.64 (95% CI: 0.53–0.73) and 0.75 (0.63–0.84), respectively. For the well-trained raters, they were 0.70 (0.60–0.78) and 0.82 (0.73–0.88), respectively; for the experienced raters, they were 0.81 (0.77–0.85) and 0.89 (0.80–0.94), respectively. The inter-rater versus weighted agreement coefficients for all the raters were 0.43 (0.28–0.58) versus 0.48 (0.29–0.67). For the well-trained raters, they were 0.50 (0.38–0.61) versus 0.61 (0.49–0.72), and for the experienced raters, they were 0.67 (0.50–0.85) versus 0.79 (0.64–0.94). Full understanding and experience led to higher reliability. Use of the algorithm can help standardize Schroth exercise treatment.

## 1. Introduction

Adolescent idiopathic scoliosis is a three-dimensional deformity of the spine affecting 2–3% of teenagers who are mostly female [1]. Scoliosis can lead to chronic consequences, including external torso deformity, pain, limited function, and poor self-image [2]. Preventing curves from progressing before maturity is important to: (1) reduce the risk of consequences described above, which typically manifest once curves exceed 50 degrees; and (2) limit or fully prevent progression during adulthood.

Schroth exercises are physiotherapeutic scoliosis-specific exercises that aim to correct posture and curvatures by improving the endurance and control of the muscles affected by scoliosis [3]. Recent systematic reviews have demonstrated that Schroth exercises can improve scoliosis curves [4,5,6], self-image [5,7], and back muscle endurance [5,7], while also decreasing pain [5,7]. A recent survey of Scoliosis Research Society members indicated that 88% support funding scoliosis-specific exercise research, and a growing number prescribe such exercises [8]. In fact, the use of Schroth exercises has increased over the past decade due to the greater interest of patients and families [8].

Schroth therapists use a classification system [9] to prescribe exercise treatment for patients with scoliosis that is adapted to each patient’s curve type. During a clinical assessment, the influence of scoliosis on the alignment and posture of each of the four body blocks is visually appraised to determine the curve pattern. The four blocks are: (a) hip-pelvic block, representing pelvis and lower extremities; (b) lumbar block, representing the part of the spine where the lumbar curve appears; (c) thoracic block, where the thoracic curve appears; and (d) shoulder block, including the shoulders and the neck (Figure 1). Scoliosis presentation can be classified into four Schroth curve types (Figure 1), two thoracic (3c and 3cp) and two thoracolumbar/lumbar (4c and 4cp) patterns. The reliability of the Schroth classification system is unknown.

From left to right, the first illustration shows a body without scoliosis; 3c is a major thoracic curve, with a small or no lumbar curve and a balanced pelvis; 3cp is a single long thoracic curve with the unbalanced pelvis deviated in the opposite direction of the thoracic convexity; 4c is a major lumbar curve compensated by a thoracic curve of similar importance (not necessarily in terms of Cobb angles) and a balanced pelvis; 4cp presents a major thoraco-lumbar or lumbar curve, with or without a thoracic curve, and with the unbalanced pelvis shifted in a direction opposite to the lumbar convexity. The plus underneath each curve representation signifies the side on which the major curve convexity appears [10].

Rigo et al. developed a classification to guide the design of the Cheneau rigid braces [11]. Rigo’s classification is based on the Schroth classification, but it distinguishes five curve types based on the location of external features and the position of the pelvis. Radiological assessment is used to further subclassify the curves. The Rigo classification has acceptable intra- and inter-rater reliability, with a kappa of 0.87 and ranging from 0.61 to 0.81, respectively [11]. Weiss also developed a classification to guide the Cheneau Light bracing, which, similarly, relies on the Schroth classification [12]. Weiss further differentiated between the curve patterns where the pelvis is imbalanced to create two additional categories, but the reliability is unknown. Several other classifications to guide treatment for scoliosis exist, including King’s [13], Lenke’s [14], and Peking Union Medical College’s classifications [15]. All aim to help plan surgery, with King’s also used to guide bracing. Although the Peking Union Medical College’s classification was found to be more reliable (kappa = 0.90) [15] than King’s (0.64) and Lenke’s classifications (0.73) [16], Lenke’s has been endorsed by the Scoliosis Research Society and is used most widely.

Although these reliable scoliosis classifications exist, they are used to inform bracing or surgical treatments rather than exercise therapy, and they rely on radiographs. Conversely, the Schroth classification mainly relies on clinical assessment to guide exercise treatment. This is consistent with most Schroth therapists not having direct access to radiographs. A reliable curve classification strategy is needed to help standardize Schroth exercise treatment delivery. Therefore, for the Multicenter Schroth exercise trial for scoliosis (NCT01610908) [6,7,17], we proposed a rule-based algorithm to assist therapists in classifying patients. Algorithms have previously been used successfully to maximize the reliability of the Cobb angle measurements and the King’s classification of patients with scoliosis, for surgery purposes [18].

To demonstrate the generalizability of the novel classification algorithm, it was important to involve varied therapists, representative of the population of certified Schroth therapists worldwide. Although Schroth therapists are spread internationally, the majority were trained by the same instructor in Germany and could, thus, be contacted from the institution’s registry. It was not feasible to have many raters test the same patients in person due to travel requirements and patient fatigue. However, because the curve classification assessment is simple, a reliability study was conducted by presenting videos of the same patients. The benefits of video recordings are that they allow for (1) blinded ratings, (2) a larger sample of widely distributed raters to be involved to maximize generalizability, and (3) minimizing the patients’ burden due to repeated evaluations.

Given these considerations, the purposes of this study were to: (1) propose a Schroth classification algorithm; and (2) determine the intra- and inter-rater reliability of Schroth therapists in classifying adolescents with idiopathic scoliosis using the algorithm. We hypothesized that therapists experienced with the algorithm and those reporting that they fully understood the classification algorithm training material would achieve adequate reliability.

## 2. Methods

### 2.1. Research Design

In this intra- and inter-rater reliability study, therapists independently rated videos twice, with at least one week in between assessments.

### 2.2. Study Participants and Raters

Forty-four volunteers with scoliosis were consecutively recruited from a specialized scoliosis clinic within a university hospital. The inclusion criteria were: (1) diagnosis of adolescent idiopathic scoliosis; (2) all curve types; (3) 10 to 18 years old; and (4) curves between 10 and 50°. Other scoliosis diagnoses and/or a history of scoliosis surgery were ineligible. All therapists trained by the German Schroth clinic and fluent in English were invited to participate as raters in this study. The first 10 therapists to volunteer were enrolled. All participants and raters gave informed consent to participate. The study was approved by the Health Research Ethics Board—Health Panel at the University of Alberta (Pro00019547).

### 2.3. Physical Examination

Participants were videotaped while standing in habitual posture presenting their front, left, right, and back sides for 10 s each, and while performing the Adam’s forward bending test (≈20 s). Maximal thoracic and lumbar rotation measurements were obtained using a scoliometer. High intra- (ICC = 0.92) and inter-rater (0.89) reliability were reported for these measurements [19]. Video capture was standardized using drapes and a frame holding a posture assessment grid to ensure consistent lighting and to aid the visualizing of postural features. A plumb line was aligned with the umbilicus in the front, the gluteal cleft for the back, or the lateral malleoli for the side views. The participants’ identity was hidden using iMovie (v8.0.6 Apple, Cupertino, CA, USA), and the scoliometer measurements were overlaid in the videos. Edited videos lasted about 60 s. Figure 2 presents a representative 3cp curve type. Other curve types are presented online (Appendix A Presentation of a representative case with 3c, 4c, and 4cp curve types, respectively).

### 2.4. Classification Algorithm

The Schroth classification algorithm and operational definitions were designed based on the 2011 Schroth training manual by the consensus of two experienced Schroth therapists with research experience [9]. The international Schroth instructor reviewed the material before testing. The algorithm guides therapists in detecting key postural features to determine the Schroth curve type (Figure 3). Each algorithm step leads to a “yes/no” decision, with the final step resulting, unambiguously, in selecting one Schroth curve type. To assist therapists in classifying, operational definitions and instructions were provided (Appendix A: Operational definitions for Schroth Classification Algorithm).

The algorithm starts with the evaluation of the pelvis displacement relative to midline, with a displaced pelvis being considered unbalanced (Figure 3). If the pelvis is found unbalanced, the relationship between the pelvis and lumbar spine is assessed. If the lumbar spine and the pelvis block deviate in the same direction (i.e., are “coupled”), the classification is 3cp. If the lumbar and pelvis blocks deviate in opposite directions (i.e., are “uncoupled”), the classification is 4cp. If the pelvis is balanced, therapists determine whether it is uncoupled from the lumbar spine and if a prominent hip is observed. In this case, therapists assess the relative importance of the thoracic prominence (rib hump) and of the lumbar prominence. If the thoracic prominence is judged to be more significant, the classification is 3c. Otherwise, the classification is 4c.

### 2.5. Rater Training

Each therapist-rater was given secure access to a website, where the algorithm, user’s guide, and operational definitions could be downloaded. After reviewing this material, raters streamed four videos from the website representing each of the Schroth curve types. After viewing each video, therapists provided decisions for each algorithm branch and the final classification and were encouraged to note any comments. The primary author of this study reviewed the comments and prepared explanatory videos to highlight the algorithm decisions for each of the four Schroth curve types in reference to the relevant section of the instructions.

### 2.6. Rating Procedures

After training, 44 patient videos were streamed in a random order on the website. Therapists could view the videos as often as needed. After the first round of rating was completed, access was blocked for a week. Subsequently, the same 44 videos were streamed again in a different random order. Therapists were blinded to their first ratings, the participants’ identity, and other therapists’ ratings.

### 2.7. Analyses

#### 2.7.1. Gwet’s Agreement Coefficient

Because of kappa’s well-documented limitations [20,21], we used Gwet’s agreement coefficient (AC1) [20] to determine intra- and inter-rater reliability as well as weighted reliability [20]. Percentage of agreement is also reported with 95% confidence intervals [20]. Reliability estimates were obtained for: the entire sample of raters; a subgroup of two therapists who conceptualized and had used the algorithm in an ongoing trial (labeled “experienced”); and a subgroup of six therapists who self-reported full understanding of the algorithm (labeled “well-trained”). Calculations were performed using AgreeStat 2013.1 for Excel Windows (Advanced Analytics, Gaithersburg, Maryland, USA).

We used Gwet’s benchmarking method, which focuses only on the part of the estimated agreement coefficient that confidently reflects genuine and not chance agreement [20]. This “significant part” was obtained by subtracting Gwet’s “critical value”, determined based on the number of participants, raters, and rating categories, from the estimated agreement coefficient [20]. With 44 participants, 10 raters, and 4 classification options, the critical value was 0.08 [20]. Reliability estimates between 0.61 and 0.80 were, thus, considered substantial. Adequate reliability to recommend clinical use was set a priori at ≥0.61. Mean reliability coefficients were calculated using Fisher’s transformation.

#### 2.7.2. Weighted Agreement

A weighted analysis was justified because differences in Schroth therapies have greater significance when misclassifying between some categories than others. Because the 3c and 4c curve patterns have a balanced pelvis, the exercise prescription will not differ as much as between the 3cp versus 4cp (major thoracic versus major lumbar with imbalanced pelvis compensating for different major curves directions), 4cp versus 3c (major lumbar with imbalanced pelvis versus thoracic major curve with balanced pelvis), or the 3cp versus 4c curve patterns (major single thoracic curve with imbalanced pelvis versus major lumbar curve with thoracic curve compensation and balanced pelvis). For 3c versus 3cp and for 4c versus 4cp patterns, the differences in therapy would be only in the prescription of exercises related to the imbalanced pelvis.

Pairings involving classifications where the treatment would occur in opposite directions (e.g., pelvis correction in 3cp versus 4cp) or focus on a different part of the posture (e.g., in 3c versus 4cp) were assigned no agreement (0.00). Full agreement weighting (1.00) was assigned to same-group ratings. The weights for each pairing, including those representing partial agreement determined a priori, are summarized in Table 1.

#### 2.7.3. Sample Size

Based on Gwet’s recommendations [20] of tolerating ≤15% error margin on the coefficient of variation of the percent agreement (standard error/percent agreement) and 20% relative error on the percent agreement (width of 95% confidence interval around the percent agreement), 44 volunteers with scoliosis and 10 raters were required for 2 repeated evaluations with 4 classification options [20].

## 3. Results

### 3.1. Participants and Therapists

The participants’ characteristics are shown in Table 2. Based on the first experienced rater’s ratings, there were nine 3c, twelve 3cp, six 4c, and seventeen 4cp Schroth curve types. Sixty-five international therapists were invited to participate, sixteen consented, and the first six to complete the ratings, in addition to the three research therapists and the Schroth instructor, were included. The age of the six female and four male therapists (mean ± standard deviation) was 46.7 ± 12.3 years old. At the time of rating, they had worked for 6.6 ± 6.4 years as Schroth therapists (range: 3 to 23 years). Five of the therapists had treated more than 75 patients in their career, two between 26 and 75, two between 11 and 25, and one between 4 and 10 patients. The therapists’ self-reported understanding of the classification algorithm ranged from 50% to 100%. Six therapists reported 100% understanding. All therapists found the algorithm useful and user-friendly, and 7 out of 10 stated they would continue using it clinically.

### 3.2. Intra-Rater and Weighted AC1 Agreement Coefficients

The overall mean intra-rater agreement coefficient was 0.64 (95% confidence interval: 0.53–0.73), and the mean weighted intra-rater coefficient was 0.75 (0.63–0.84).(Table 3) Experienced raters reached higher estimates with a mean agreement coefficient of 0.81 (0.77–0.85) and weighted coefficients of 0.89 (0.80–0.94). The six well-trained therapists reached a mean intra-rater coefficient of 0.70 (0.60–0.78) and a weighted intra-rater coefficient of 0.82 (0.73–0.88).

### 3.3. Inter-Rater and Weighted AC1 Agreement Coefficients

The overall mean inter-rater agreement coefficient was 0.43 (95% confidence interval 0.28–0.58). (Table 4) The corresponding value for the experienced raters was 0.67 (0.50–0.85). The well-trained therapists’ inter-rater coefficient was 0.50 (0.38–0.61). Overall, the weighted coefficient was 0.48 (0.29–0.67). The experienced and well-trained raters had weighted coefficients of 0.79 (0.64–0.94) and 0.61 (0.49–0.72), respectively.

## 4. Discussion

In this study, we proposed a standardized algorithm to achieve adequate rater’s reliability in classifying the Schroth curve type. This algorithm was developed to assist with standardizing exercise prescription as part of the Multicentre Schroth Exercise Trial for Scoliosis [6,7,17]. For the first time, we presented the intra- and inter-rater reliability of Schroth exercise therapists in classifying patients with idiopathic scoliosis treated conservatively. The overall weighted intra-rater, intra-rater, and weighted agreement coefficients for the experienced and well-trained therapists, as well as the weighted inter-rater coefficients for the experienced therapists, met our a priori threshold (≥0.61) for recommending clinical use. Note that the weighted reliability analysis is the most relevant one for Schroth clinical practice because some categories share similar postural defects and, thus, similar exercises, whereas others do not share such postural or exercise prescription similarities.

Rigo’s [11] and Weiss’ [12] classifications were derived from the Schroth curve classification to guide brace design. The King’s [13], Lenke [14], and Peking University Medical Center [15] classifications were proposed to guide surgical procedures and, hence, are rather tailored to more or significant deformities. Therefore, comparisons with other reliability classification studies are difficult because the Schroth classification tested for reliability in the present study is tailored for the classification of patients with small curves for a different purpose: selecting exercises.

With the usage of an algorithm, the intra-rater reliability of King’s classification improved from a previously published kappa of 0.64 [21] to 0.85[17] and the inter-rater reliability from 0.44 [22] to 0.82 [18]. Similarly, when compared with the gold standard—software automatically classifying Lenke curve types according to radiographic measurements of Cobb angles—the accuracy of Lenke’s classification improved from 0.77 to 0.93 by using an algorithm [23].

To minimize differences in therapists’ interpretation of the rating procedure, we proposed a standardized Schroth classification algorithm with operational definitions and instructions. Still, the two therapists experienced in using the algorithm had the highest reliability. Overall, the self-reported comprehension of the algorithm ranged between 50% and 100% and significantly correlated with the intra-rater reliability coefficient in this sample of 10 therapists (Spearman’s rho = 0.68, *p* = 0.03). Additional training may be needed for therapists with a lower level of understanding. Furthermore, modifications of the algorithm could be considered to improve its reliability.

To avoid sampling bias and maximize generalizability, study participants were consecutively selected from our specialized scoliosis clinic. However, Lenke 1, 2, and 5 curve types that partially correspond to the 3c, 3cp, and 4cp curve types, according to Schroth, reportedly account for 83% of all scoliosis patterns [24]. Thus, underrepresentation of the 4c Schroth curve type might have been observed.

After training, a professional photographer helped improve the quality of the videos by minimizing shadows affecting the three-dimensional perception of the postural characteristics of scoliosis. Most therapists reported that the quality of the videos was adequate. During treatment, a therapist, over time, could confirm or change classification and adjust the treatment if needed. Our results may therefore reflect a lower limit of the reliability because therapists were shown only one assessment over which they had no control.

Although video assessments have been successfully used in reliability studies [25,26,27], the inability to rate scoliotic presentation in person might have influenced the estimates. Reliability may be overestimated by using a video presentation that removed the errors that could be caused by therapists providing different instructions during the evaluation. On the other hand, reliability may be underestimated if issues related to observing evaluation videos caused more difficulty in classifying than if the exams were conducted in person. Difficult-to-rate patients may be better assessed if therapists could evaluate them while using different postures or while performing movements other than forward bending. Reliability may be improved if this flexibility in the testing procedure led to a better classification accuracy; however, it could also lead to poorer reliability if it increased the variability in the testing procedures. Typically, a therapist observes a patient from each side and performs the Adam’s bending test to assess asymmetries. As such, we think that the current standardized video assessment adequately reflects the typical assessment of most practicing Schroth therapists.

### Limitations and Future Work

While adequate to meet our power calculations a priori, the samples of 44 patients, and 10 raters (including 2 experienced and 6 well-trained raters) is relatively small to ensure generalizability for all other raters and patients. Replicating this study with additional patients and raters would help demonstrate generalizability. To assess the true value of the algorithm, reliability should be compared with the classification reliability observed before therapists are trained in using the algorithm. In the future, the sources of disagreements should be determined by calculating the reliability at each decision step and identifying the decision steps most responsible for the classification error. If there are some patients who are difficult to rate, their characteristics could be assessed. The algorithm could also be tested with therapists assessing patients in person. Finally, it would be important to assess whether the reliability changes with the severity of the scoliosis.

## 5. Conclusions

We proposed a simple algorithm to maximize the therapists’ reliability in classifying Schroth curve type and assist in the prevention of treatment errors due to misclassification. The algorithm was well accepted, and most raters reported that they were planning to use it clinically. Experienced and well-trained raters achieved adequate reliability. Our results suggest that the algorithm can be used clinically with sufficient training [28].

## Figures and Tables

**Figure 1 children-10-00523-f001:**
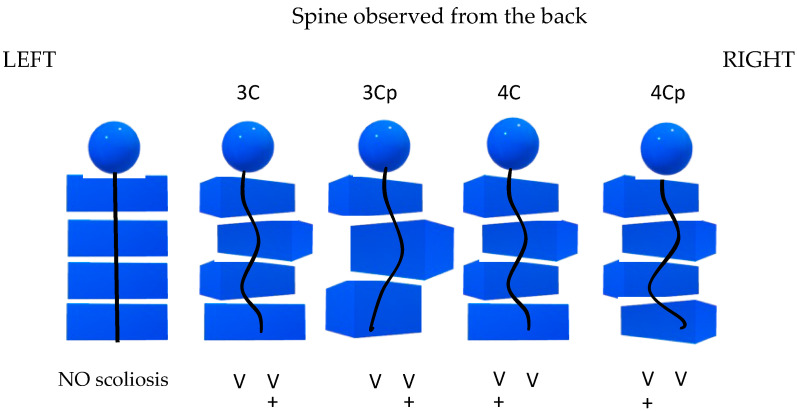
Schroth classification illustrated based on the most common right-convex thoracic or left lumbar curve direction.

**Figure 2 children-10-00523-f002:**
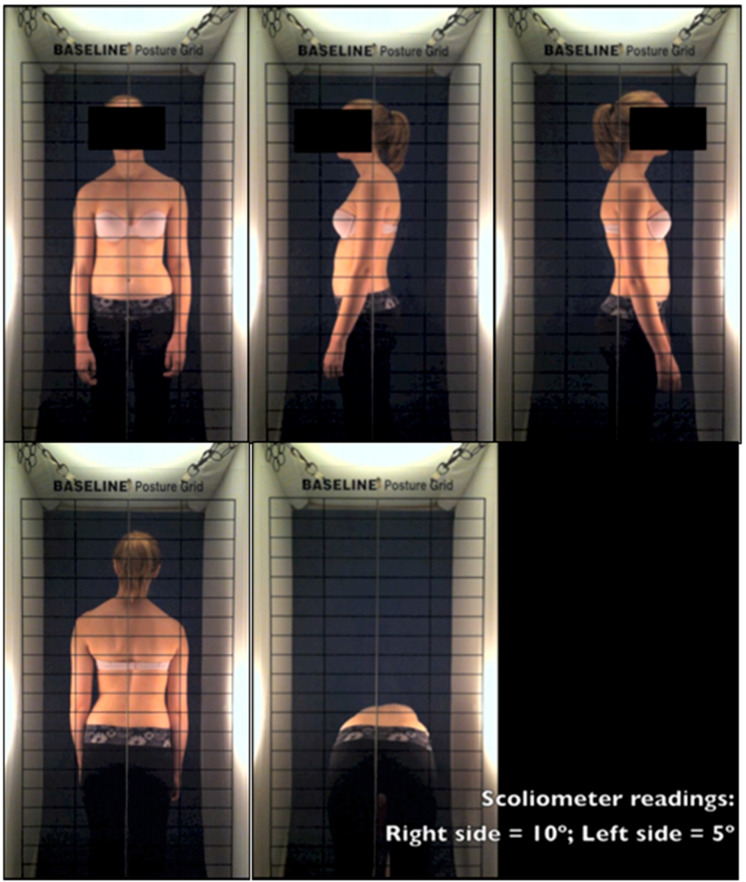
Snapshots from the video assessment of a 3cp curve type. The plumb line was aligned with the umbilicus, left malleolus, right malleolus, and the gluteal cleft in the front, left, right, and back views, respectively. The Adam’s bending test video clip displayed scoliometer readings for the largest rotation to the right and left, measured over the trunk.

**Figure 3 children-10-00523-f003:**
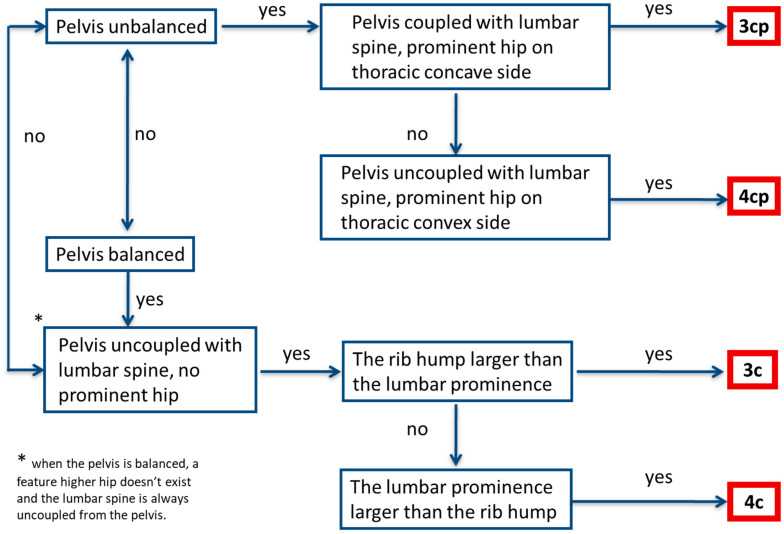
Schroth-curve-type classification algorithm.

**Table 1 children-10-00523-t001:** Partial agreement weights.

	3c	3cp	4c	4cp
3c	1.00	0.75	0.50	0.00
3cp	0.75	1.00	0.00	0.00
4c	0.50	0.00	1.00	0.75
4cp	0.00	0.00	0.75	1.00

**Table 2 children-10-00523-t002:** Description of the participants with adolescent idiopathic scoliosis.

	N	Mean	Standard Deviation	Minimum	Maximum
Age (years)	44	14.2	2.0	10.0	18.0
Upper thoracic Cobb angle (°)	8	25.0	7.1	15.0	38.0
Major thoracic Cobb angle (°)	36	26.7	10.3	10.0	49.0
Thoracolumbar/lumbar Cobb angle (°)	37	25.8	10.0	10.0	48.0

**Table 3 children-10-00523-t003:** Intra-rater AC1 and weighted AC1 coefficients and percent agreement with 95% confidence intervals.

	AC1 (95% CI)	Percent Agreement (95% CI)	Weighted AC1 (95% CI)	Weighted Percent Agreement (95% CI)
Experienced 1	0.79 (0.64–0.94)	84.1 (73.0–95.2)	0.92 (0.87–0.98)	96.0 (93.2–98.9)
Experienced 2	0.83 (0.69–0.96)	86.3 (75.9–96.8)	0.85 (0.71–0.99)	91.5 (83.7–99.2)
Rater 1	0.47 (0.27–0.67)	59.1 (44.1–74.0)	0.60 (0.41–0.79)	78.4 (68.5–88.3)
Rater 2	0.52 (0.32–0.71)	63.6 (49.0–78.3)	0.58 (0.36–0.79)	78.4 (67.5–89.4)
Rater 3	0.73 (0.57–0.89)	79.5 (67.3–91.8)	0.89 (0.82–0.96)	94.3 (90.7–97.9)
Rater 4	0.51 (0.30–0.71)	61.5 (45.8–77.2)	0.57 (0.35–0.80)	76.3 (64.2–88.4)
Rater 5	0.61 (0.43–0.80)	70.5 (56.6–84.3)	0.68 (0.50–0.87)	83.0 (73.1–92.8)
Rater 6	0.34 (0.14–0.54)	48.8 (33.5–64.2)	0.41 (0.18–0.64)	67.4 (54.8–80.1)
Rater 7	0.72 (0.55–0.88)	77.2 (64.5–90.0)	0.80 (0.66–0.95)	88.0 (79.9–96.2)
Rater 8	0.64 (0.46–0.82)	72.7 (59.2–86.3)	0.83 (0.72–0.94)	90.9 (85.3–96.5)
*Mean (overall)*	*0.64 (0.53–0.73)*	*72.0 (63.9–78.6)*	*0.75 (0.63–0.84)*	*86.7 (79.9–91.3)*
*Mean (experienced)*	*0.81 (0.77–0.85)*	*85.0 (82.9–86.9)*	*0.89 (0.80–0.94*	*94.0 (86.8–97.3)*
*Mean (well-trained)*	*0.70 (0.60–0.78)*	*76.6 (69.4–82.3)*	*0.82 (0.73–0.88)*	*90.1 (84.8–93.6)*

**Table 4 children-10-00523-t004:** Inter-rater AC1 and weighted AC1 coefficients and percent agreement with 95% confidence intervals.

	AC1 (95% CI)	Percent Agreement (95% CI)	Weighted AC1 (95% CI)	Weighted Percent Agreement (95% CI)
All raters (N = 10)	0.43 (0.28–0.58)	56 (45–67)	0.48 (0.29–0.67)	73 (63–82)
Experienced raters (N = 2)	0.67 (0.50–0.85)	75 (62–88)	0.79 (0.64–0.94)	89 (81–96)
Well-trained raters (N = 6)	0.50 (0.38–0.61)	61 (53–70)	0.61 (0.49–0.72)	79 (73–85)

## Data Availability

The data presented in this study are available in Appendix A.

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
