# Peer review of "Algorithm for Schroth-Curve-Type Classification of Adolescent Idiopathic Scoliosis: An Intra- and Inter-Rater Reliability Study†"

_children, 2023, doi:10.3390/children10030523_

Round 1

Reviewer 1 Report

line 62 - there is a lack of reference

reference nr 7 and 16 are the same paper

why do data from tables 2 and 3 are not the same? title and parameters are the same, but there are differences in values.

small study group and small experienced therapists group are the paper's weak points, which should be mentioned. 

Author Response

See our file uploaded. 

Please not this response also includes our response to comments from the editor shared when we received reviewer 1 comments. 

Reviewer 2 Report

This is a well-written, conceived and conducted study.  The rationale is clearly articulated and limitations recognized.  Implications and suggestions for future research are stated.  No suggestions for improvements are needed. 

Author Response

Thank you for your positive review. We have uploaded our response below. 

Reviewer 3 Report

This manuscript by Sanja Schreiber et al presented a clinical study to develop a new Schroth classification algorithm and tested the reliability using clinical patients as subject and therapist as rater. The study is important to standardize classifications of idiopathic scoliosis and guided their exercise therapy. The study is well designed and implemented study and the manuscript is well written. Recommend to accept in current form. 

One comment:

Lines 254-255, please make it more clear for the following sentence:

“Therefore, comparisons with other reliability classification reliability studies are difficult.”.

Author Response

See our response letter uploaded. 
